# Choriocapillaris flow deficit in a pachychoroid spectrum disease using en face optical coherence tomography angiography averaging

Miho Tagawa[1], Sotaro Ooto[1]*, Kenji Yamashiro[1,2], Hiroshi Tamura[1], Akio Oishi[1], Akihito Uji[1], Manabu Miyata[1], Masahiro Miyake[1], Ayako Takahashi[1], Ai Ichioka[1], Akitaka Tsujikawa[1]

1 Department of Ophthalmology and Visual Sciences, Kyoto University Graduate School of Medicine, Kyoto, Japan, 2 Japanese Red Cross Otsu Hospital, Otsu, Japan

* ohoto@kuhp.kyoto-u.ac.jp

**Data Availability Statement:** All relevant data are within the paper and its Supporting Information files.

## Abstract

### Purpose

To investigate the choriocapillaris changes associated with pachychoroid pigment epitheliopathy (PPE) in comparison with healthy eyes.

### Methods

Nine 3 × 3 mm macular optical coherence tomography angiography images were acquired in patients with PPE and age-matched healthy participants. Multiple en face image averaging of the choriocapillaris was binarized for quantitative image analysis of the flow voids. In PPE eyes, we evaluated the presence of pachyvessels and the association between the location of the choriocapillaris flow deficit and pachyvessels.

### Results

Thirty-two eyes with PPE and 30 eyes of healthy participants were included. In PPE eyes, the mean total area (1.16 ± 0.18 vs. 0.91 ± 0.16, p < 0.001) and average size of the flow voids (790 ± 144 vs. 520 ± 138; p < 0.001) were significantly larger than those in control eyes. Composite images of the choriocapillaris and choroid showed choriocapillaris flow deficits just above and outside the pachyvessels. The mean proportion of the flow void area overlying the pachyvessels against the whole flow void area of the choriocapillaris was 21.3% ± 10.2% (9.38%-44.42%) in PPE eyes.

### Conclusions

In PPE eyes, the blood flow area of the choriocapillaris decreased diffusely within the macular area compared to control eyes, and the choriocapillaris flow deficit was not necessarily related to pachyvessel location.

**Funding:** This study was supported in part by the Japan Society for the Promotion of Science (JSPS), Tokyo, Japan (Grant-in-Aid for Scientific Research, no. 16K11321). The funding body had no role in the design or conduct of the study, management, analysis, and interpretation of the data, or the preparation, review, or approval of the manuscript.

**Competing interests:** The authors have declared that no competing interests exist.

## Introduction

Pachychoroid spectrum diseases have recently been reported as a new clinical entity that includes conditions such as central serous chorioretinopathy (CSC), pachychoroid pigment epitheliopathy (PPE), polypoidal choroidal vasculopathy (PCV), pachychoroid neovasculopathy (PNV), and pachychoroid geographic atrophy (GA) [1–7]. Late-stage complications such as chronic CSC or progression into advanced stages such as PCV, PNV, and pachychoroid GA may cause severe visual disturbances. Although several studies have reported the effectiveness of anti-vascular endothelial growth factor (VEGF) therapy for PNV [5, 6, 8–10], high-level evidence for this treatment has not been established. Despite anti-VEGF treatment, the prognosis is usually poor. Moreover, no treatments for pachychoroid GA as well as GA related to age-related macular degeneration (AMD) have been described to date, and natural course of visual loss has been previously reported [11, 12]. One of the reasons for this lack of treatment options is that the pathogenesis of pachychoroid spectrum diseases is poorly understood.

Among pachychoroid spectrum diseases, PPE is thought to represent the earliest clinical stage. PPE is characterized by retinal pigment epithelium (RPE) abnormalities overlying areas of choroidal thickening without subretinal fluid [1]. Elucidation of the pathology associated with PPE can improve our comprehension of the common mechanisms underlying pachychoroid spectrum diseases. In these diseases, a key feature is inner choroidal attenuation described by focal or diffuse attenuation of the choriocapillaris [3, 13, 14]. Dansingani KK et al. used swept-source optical coherence tomography (SS-OCT) and observed increased choroidal thickness and dilated outer choroidal vessels in en face images of eyes with pachychoroid spectrum disease [3]. Gal-Or O et al. used OCT angiography to show the relation between the flow signal attenuation with reduced inner choroidal thickness and pachyvessels in pachychoroid disease [14], suggesting the involvement of choriocapillaris abnormalities.

Optical coherence tomography angiography (OCTA) contributes to visualize the depth-resolved image of the retinal and choroidal microvasculature through motion contrast of blood flow. It is a noninvasive imaging modality that does not require dye injections, avoiding the problems associated with dye leakage, and offers a depth resolution adequate for selection of the choriocapillaris layer. However, choriocapillaris imaging using current OCTA technology is still challenging because of the complex microstructure of the choriocapillaris. Several studies have recently reported that averaging of multiple en face OCTA images improved the image quality of the retinal vasculature and the choriocapillaris by reducing speckle noise and compensating for the inadequate signals of a single image, thereby significantly improving quantitative measurements [15, 16].

SS-OCT at longer wavelengths is characterized by a high-speed scan rate and high penetration in comparison with spectral-domain OCT, and it makes it possible to capture of high-contrast images of the choroid. Averaging of multiple en face SS-OCTA images enables visualization of the meshwork structure of the choriocapillaris in vivo, and is suitable for quantitative assessments of the choriocapillaris microstructure [17]. Using this averaged en face images of choriocapillaris improves quantifying and analyzing the number and area of flow voids [16, 18] (areas without flow signals), and flow deficit (increased flow void area).

The purpose of the present study was to examine the PPE-associated changes in the choriocapillaris in comparison with healthy eyes by using multiple en face OCTA averaging, and to evaluate the relationship between the flow void area of the choriocapillaris and the location of pachyvessels in eyes with PPE.

The vessels of Haller's layer from the macular region drain into the upper temporal vortex vein, while those in the lower region flow into the lower temporal vortex vein.

Therefore, these vessels crossing the macular region also defined as 'natural oblique vessels,' represented terminal tributaries of the vortex vein. Recently Matsumoto et al. reported that a

horizontal watershed passing through the macula which divide superior and inferior macular vortex veins was seen in 80% of normal eyes, while venous anastomosis between the superior and inferior vortex veins was detected in 20% of normal eyes. They also showed the horizonal watershed had disappeared and collateral veins had developed via anastomosis between the superior and inferior vortex veins in 90% of eyes with PNV [19]. Thus, pathological involvement of the vortex vein's terminal tributaries ideally should be evaluated in wide field images. In this study we defined pachyvessels as large outer choroidal vessels coming from a dilated vortex ampulla with a wide-field image of ICGA and wide-field en face OCT images. However, choriochapillaris imaging can only be applied within the central 3 × 3 mm area due to the lateral resolution of OCTA imaging, therefore the analysis of this study was limited to the foveal and parafoveal regions.

## Methods

This study was approved by the ethics committee at Kyoto University Graduate School of Medicine approved and conducted according with the tenets of the Declaration of Helsinki. We obtained written informed consent from each participant before any study procedures or examinations.

### Participants

The current study was a prospective cross-sectional study. We enrolled patients with PPE who visited the Macular Service at Kyoto University Hospital and age-matched heathy participants. Eyes of PPE were fellow eyes of PNV, CSC or PPE. If both eyes were eligible, the right eye was included. We recruited healthy eyes of healthy volunteers and unaffected fellow eyes of patients with unilateral retinal diseases (e.g., epiretinal membrane, vitreomacular traction syndrome) [16]. In the current study, subjects with the systemic conditions like diabetes mellitus or those with poorly controlled hypertension were excluded. Each participant conducted an ocular comprehensive examination, which included color fundus photography (Topcon, Tokyo, Japan), axial length measurement using ocular biometry (IOLMaster; Carl Zeiss Meditec, Jena, Germany), spectral-domain OCT by Spectralis (Heidelberg Engineering, Heidelberg, Germany), and swept-source OCTA (PLEX Elite 9000; Carl Zeiss Meditec, Dublin, CA, USA). All patients with PPE also underwent fundus autofluorescence (FAF) imaging, fluorescein angiography (FA), and indocyanine green angiography (IA) by the Spectralis HRA+OCT system (Heidelberg Engineering, Heidelberg, Germany).

By using multimodal images, PPE was diagnosed by the presence of pachychoroid features and RPE abnormalities without subretinal fluid. RPE abnormalities were characterized by granular hypo-autofluorescence on FAF images. Pachychoroid features were defined as reduced fundus tessellation on color fundus photographs, pathologically dilated outer choroidal vessels on OCT and IA images [20], and/or regional choroidal vascular hyperpermeability (CVH) on IA images. Pachyvessels were delineated as pathologically dilated outer choroidal vessels with attenuation of the choriocapillaris on OCT [20]. On wide-field IA and wide-field (12mm × 12mm) en face OCT images, pachyvessels were found to expand from one or more vortex vein ampullas [21]. The existence of dilated choroidal vessels was assessed by using multimodal imaging. The diagnoses were made by two retinal specialists; in cases involving a disagreement in the diagnosis, a senior retinal specialist determined the final diagnosis.

### Measurement of choroidal thickness

We delineated choroidal thickness as the distance from the Bruch membrane to the choriosceleral interface. We manually measured the choroidal thicknesses at the center of the fovea from vertical and horizontal scans by using a built-in caliber tool and averaged each thickness.

## Multiple en face OCTA image averaging

We evaluated macular OCTA images acquired from all patients by using a PLEX Elite 9000 unit that uses a swept laser source with a central wavelength of 1050 nm and operates at 100,000 A scans per second. Using this system, we repeatedly obtained central 3 × 3 mm scans on the fovea per eye until nine OCTA cubes with enough high quality could be obtained. The choriocapillaris was segmented as an 8-μm-thick slab stating 29 μm posterior to the RPE-Bruch membrane complex with a fully automated segmentation algorithm using built-in software [15, 16]. We averaged multiple en face image of the choriocapillaris by using OCTA images of the superficial capillary plexus for registration, and this same transformation information was then adjusted to the choriocapillaris layer [15, 16]. The score of signal strength is displayed for each captured image, and we used the images with the score more than 8/10 for averaging and did not adopt the image with reflected artifact by blinking or poor fixation.

## Quantitative image analysis of the choriocapillaris OCTA images

We binarized the choriocapillaris OCTA images by using the Phansalkar method [22–24] to quantify and analyze the flow voids. Then, using Image J software, the threshold images were analyzed to evaluate the number of flow voids, total flow void area, and average size of the flow voids [15, 16].

## Choroidal en face OCT image

In the original segmentation by the built-in software, the choroid was segmented as a 51-μm thick slab starting 64 μm posterior to the RPE. The default segmentation of the choroid was thin and unsuitable to detect dilated outer choroidal vessels. Therefore, we manually modified the segmentation of the choroid from 64 μm posterior to the RPE to the scleral surface. Additionally, the central 12 × 12 mm en face image on the fovea was obtained per eye with PPE to evaluate the presence of pachyvessels.

## Evaluation of the presence of pachyvessels in pachychoroid pigment epitheliopathy

The presence of pachyvessels in PPE eyes was evaluated within a 3 × 3 mm area of the macula by using choroidal en face OCT images (3 × 3 mm, 12 × 12 mm) and wide-angle IA. (Fig 1) Pachyvessels were defined as dilated choroidal venous veins that continuously extended from

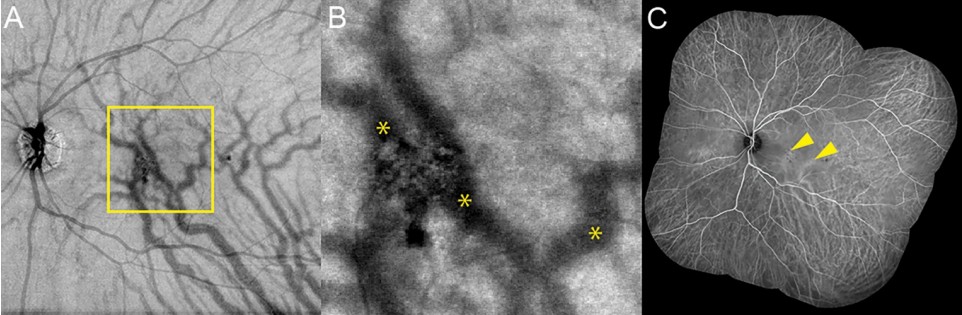

**Fig 1. Multimodal images for evaluation of the presence of the pachyvessels in the PPE eye.** (A) An en face OCT image of 12 × 12 mm at the level of the choroid. The square shows a 3 × 3 mm area covering the macula. (B) An en face OCT image of 3 × 3 mm at the level of the choroid. Asterisks show pachyvessels. (C) Panoramic images with indocyanine green angiography. The arrowheads show pachyvessels.

the dilated vortex vein, could be distinguished by their relatively larger caliber in comparison with other choroidal veins in other quadrants, and showed asymmetry of upper and lower vortex veins [3, 25, 26]. Eyes with PPE that showed pachyvessels within the 3 × 3 mm area of the macula were categorized under the "pachyvessel group". The presence of pachyvessels in each case was determined in agreement by two retinal specialists (M.T. and S.O.). Then, we divided the PPE eyes into the two groups, a pachyvessel group and a non-pachyvessel group, within the 3 × 3 mm area of the macula and compared the parameters of flow voids between the two groups.

## Measurement of the distance between RPE and choroidal venous vein

We evaluated the distance between RPE and choroidal venous vein. We manually measured the distance at the center of the fovea, 1mm superior side from the fovea, 1mm inferior side from the fovea, 1mm nasal side from the fovea, and 1mm temporal side from the fovea by using a built-in caliber tool. We used vertical and horizontal B-scans through the fovea, and averaged each distance.

## Quantitative analysis of the correlation between the location of choriocapillaris dropout and pachyvessels

To quantify the flow void areas overlying the pachyvessel, we binarized the averaged choriocapillaris en face OCTA image and choroidal en face OCT image by the Phansalkar method (radius, 15 pixels), which was a suitable design to select darker regions in potentially low-contrast images [24]. Subsequently, we merged these en face images and evaluated the percentage of the flow void area overlying the pachyvessels against the whole flow void area by using ImageJ software (version 1.51j8, available at fiji.sc, free of charge). We defined the percentage of the flow void area overlying the pachyvessels against the whole flow void area in the 3 × 3 mm area as the portion of the pachyvessels.

## Statistical analysis

We analyzed statistically by using PASW Statistics version 19.0 (SPSS, Chicago, IL). All values are presented as mean ± standard deviation. We used Chi-square tests to compare the sex ratio. Intergroup comparisons were performed using unpaired $t$-tests. Partial correlation was used for adjusting for age and sex. We used Pearson's product moment correlation coefficient to evaluate the correlation of the flow void areas overlying the pachyvessel against the entire flow void area in the choriocapillaris. A p value of $< 0.05$ was considered statistically significant.

## Results

This study included 32 eyes of 32 patients with PPE and 30 eyes of 30 healthy participants. Participant characteristics are summarized and presented in Table 1. PPE patients and healthy participants showed similar ages (62.5 ± 8.6 vs. 65.2 ± 9.9; p = 0.254), sex distributions (25 men and 7 women vs. 18 men and 12 women, p = 0.122), and axial lengths (23.99 ± 0.96 mm vs. 24.38 ± 1.14 mm; p = 0.155). However, the mean SFCT of PPE patients (374.5 μm) and significantly larger than that of the controls (p < 0.001).

The single unaveraged choriocapillaris en face images indicated a granular appearance. After averaging the findings for all eyes, the meshwork structure of the choriocapillaris clearly appeared in the averaged image (Fig 2).

**Table 1. Characteristics of PPE and control eyes.**

|  | PPE eyes (n = 32) | Control eyes (n = 30) | p value |
|---|---|---|---|
| Age (yr) | 62.5 ± 8.6 (48–82) | 65.2 ± 9.9 (40–80) | 0.254 |
| Sex (M/F) | 25/7 | 18/12 | 0.122 |
| Axial length (mm) | 23.99 ± 0.96 (22.05–26.31) | 24.38 ± 1.14 (22.36–26.86) | 0.155 |
| SFCT (μm) | 374.5 ± 81.5 (250–533) | 248.7 ± 35.3 (193–304) | < 0.001 |

PPE: pachychoroid pigment epitheliopathy; SFCT: subfoveal choroidal thickness.

Quantitative OCTA parameters of the flow voids in PPE patients and age-matched healthy participants are presented in Table 2. In PPE eyes, the mean total area (1.16 ± 0.18 vs 0.91 ± 0.16; $p < 0.001$, $p^* < 0.001$) and average size of the flow voids (790 ± 144 vs 520 ± 138; $p < 0.001$, $p^* < 0.001$) were significantly larger than those in control eyes ($p^*$ values are adjusted for age and sex). (Fig 3)

Fellow eyes of eyes with PPE were PNV: 17 eyes, CSC: 11 eyes, PPE: 4 eyes.

In PNV eyes, the mean number of flow voids were significantly smaller than those in CSC eyes (1432 ± 147 vs 1592 ± 111; $p = 0.018$). In PNV eyes, the mean average size of the flow voids was significantly larger than those in CSC eyes (842 ± 144 vs 704 ± 76; $p = 0.041$). However, there were no significant differences in the total flow void area among 3 groups.

We divided the PPE eyes into two groups, i.e., a pachyvessel group (pachyvessels located within a 3 × 3 mm macular area, n = 25) and a non-pachyvessel group (pachyvessels not located within a 3 × 3 mm macular area, n = 7), and compared these parameters between the two groups (Table 3). The two groups showed no significant differences in number of flow voids, total flow void area, average size of the flow voids, and SFCT after adjusting for age and sex. The distance between RPE and choroidal venous vein in pachyessel group was significantly shorter than that in non-pachyvessel group.

A choriocapillaris flow deficit was seen in all eyes with PPE (Fig 4). Composite images of the choriocapillaris and choroid showed that choriocapillaris deficit was seen not only just above the pachyvessels but also outside the pachyvessels. In PPE eyes with a pachyvessel within a 3 × 3 mm area (n = 25), we evaluated the quantitative measurement of the flow void area

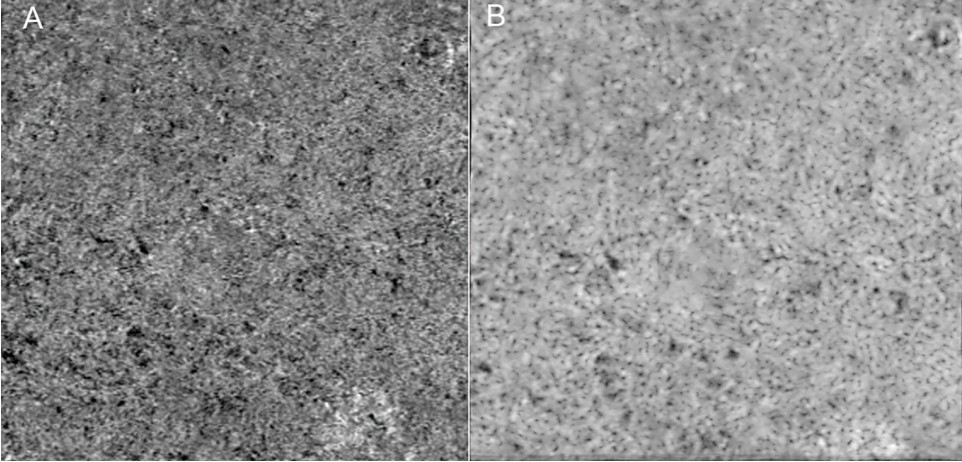

**Fig 2. Multiple en face OCTA image averaging of the choriocapillaris.** (A) An unaveraged en face OCTA image of the choriocapillaris. (B) An averaged en face OCTA image of choriocapillaris. After averaging, the complex meshwork structure of the choriocapillaris is seen clearly.

**Table 2. Quantitative OCTA parameters of the flow voids in PPE patients and age-matched healthy participants.**

|  | PPE | Control | p value | p* value |
|---|---|---|---|---|
| Number of flow voids | 1483 ± 154 (1128–1819) | 1826 ± 319 (1460–2959) | < 0.001 | < 0.001 |
| Total flow void area (mm$^2$) | 1.16 ± 0.18 (0.81–1.58) | 0.91 ± 0.16 (0.47–1.13) | < 0.001 | < 0.001 |
| Average size of the flow voids (μm$^2$) | 790 ± 144 (523–1087) | 520 ± 138 (184–727) | < 0.001 | < 0.001 |

OCTA: optical coherence tomography angiography; PPE: pachychoroid pigment epitheliopathy; VDI: vessel diameter index

p* = p values adjusted for age and sex.

overlying the pachyvessels. The mean portion of the pachyvessels in the 3 × 3 mm area was 21.0% ± 9.8% (8.03%–44.46%). PPE eyes showed a choriocapillaris flow deficit regardless of pachyvessel location. The mean proportion of the flow void area overlying the pachyvessels against the whole flow void area of the choriocapillaris was 21.3% ± 10.2% (9.38%–44.42%). There were no correlations between the portion of the flow void area overlying the pachyvessels and age, SFCT, number of flow voids, total area of flow voids, and average size of flow voids. (R = - 0.168, p = 0.422; R = - 0.322, p = 0.105; R = - 0.038, p = 0.858; R = - 0.264, p = 0.203; R = - 0.154, p = 0.462; respectively).

All data are available in the S1 Data.

## Discussion

Averaging of multiple en face SS-OCTA images enables visualization of the meshwork structure of choriocapillaris in vivo. We investigated the choriocapillaris changes associated with PPE in comparison with healthy eyes by using multiple en face OCTA imaging. The results indicated that the total area and average size of the flow voids were larger in PPE eyes than in control eyes. In PPE eyes, choriocapillaris flow deficit was seen diffusely and not necessarily located just above the pachyvessels.

In the current study, we evaluated the quantitative flow voids of the choriocapillaris and showed that the total flow void area and average size of the flow voids in PPE eyes were larger than those in control eyes. The results for flow voids in normal eyes are shown in Table 2. We examined the validity of these controls by comparing the findings with those reported

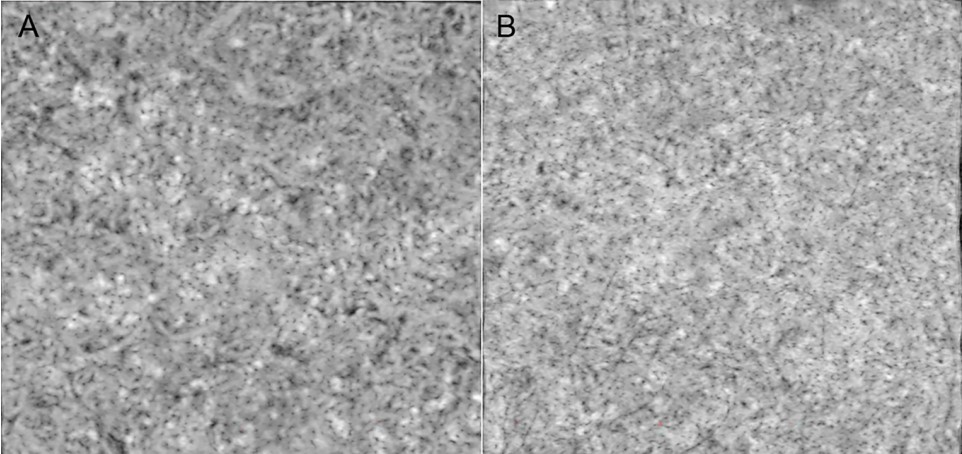

**Fig 3. Averaged choriocapillaris OCTA images of the PPE and control eyes.** (A) PPE eye. A 64-year-old man. Total area of the flow voids: 1.19 mm$^2$; average size of the flow voids: 958 μm$^2$. (B) Control eye. A 59-year-old-man. Total area of the flow voids: 0.94 mm$^2$; average size of the flow voids: 541 μm$^2$.

**Table 3. Quantitative OCTA parameters of the flow voids in PPE patients with/without pachyvessels within a 3 × 3 mm area.**

| | Pachyvessel within a 3 × 3 mm area (n = 25 eyes) | No pachyvessel within a 3 × 3 mm area (n = 7 eyes) | p* value |
|---|---|---|---|
| Age | 62.5 ± 9.0 | 62.4 ± 7.0 | 0.981 |
| Number of flow voids | 1470 ± 138 | 1528 ± 194 | 0.159 |
| Total flow void area (mm$^2$) | 1.15 ± 0.17 | 1.19 ± 0.20 | 0.137 |
| Average size of the flow voids (μm$^2$) | 792 ± 144 | 782 ± 142 | 0.243 |
| SFCT (μm) | 379.2 ± 86.2 | 358.0 ± 58.7 | 0.559 |
| Distance between PRE and choroidal venous vein (μm) | 64.0 ± 19.6 | 107.2 ± 25.9 | < 0.001 |

SFCT: subfoveal choroidal thickness

p* = p values adjusted for age and sex.

previously. Ichioka et al. reported that the mean total flow void area and average size of the flow voids in healthy eyes were 0.99 ± 0.20 mm$^2$ and 567.8 ± 201.5 μm$^2$ (81eyes; mean age, 59.5 years) using the same methods of averaged OCTA images with the current study. They also showed the age-related gains in the mean total area and size of flow voids were 4.20 × 10 mm$^2$ and 3.07 μm$^2$ per year, respectively, comparable with this study [16]. Liu K, et al. reported that in normal eyes, the mean total flow void area and average size of the flow voids were 0.3 ± 0.2 mm$^2$ and 255 ± 137 μm$^2$ by using swept-source OCTA and averaged images (23 eyes from 12 healthy participants; mean age, 30 years: range, 23–43 years) [27]. These discrepancies may be attributable to the differences in the target age group. Spaide used OCTA to show that age and hypertension affect the measurable flow characteristics of the choriocapillaris [23]. Mullins, et al. also histologically evaluated the choriocapillaris dropout and showed that it positively correlated with age [28]. Ramrattan et al. showed that capillary density of the choriocapillaris diminished in a linear fashion from approximately 0.75 in the first decade to 0.40 in the

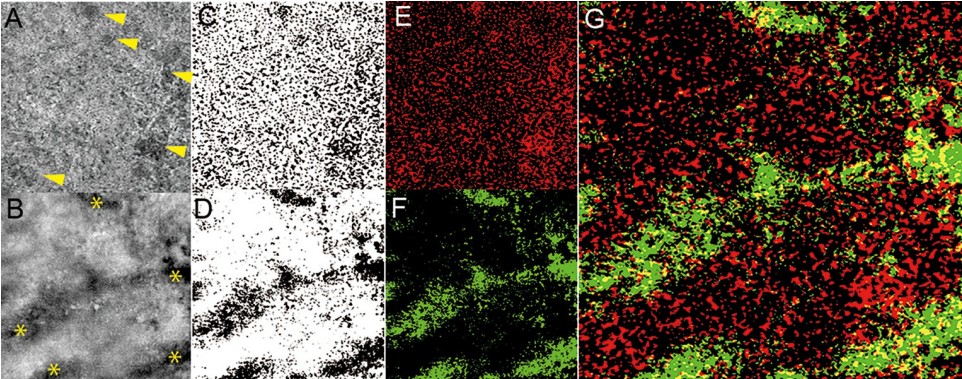

**Fig 4. Choriocapillaris flow deficit and the location of pachyvessels.** (A) An averaged en face OCTA image of the choriocapillaris within a 3 × 3 mm area. Arrowheads show the choriocapillaris flow deficit. (B) An en face OCT image of the choroid within a 3 × 3 mm area. Asterisks show pachyvessels. (C, D) These two en face images were binarized using automatic local thresholding with the Phansalkar method for the choriocapillaris slab (C) and the choroid slab (D), respectively. (E) With inversion, the black area that shows flow voids was converted to a white area and then colored in red. (F) With inversion, the black area that shows pachyvessels was converted to a white area and then colored in green. (G) Composite image of these two en face images. The flow void is red, and the pachyvessel is green. The yellow area shows the choriocapillaris flow deficit just above the pachyvessels. The red area shows the choriocapillaris flow deficit outside the pachyvessels. This composite image shows that the choriocapillaris flow deficit is seen not only just above the pachyvessels but outside pachyvessels. The portion of the flow void area overlying the pachyvessels (yellow area) against the whole flow void area of the choriocapillaris was 23.68%.

tenth decade of life by using histologic sections of normal eyes [29]. Since the mean age of the normal participants in this study was 65.2 years, we considered that the results for the normal participants were compatible with those reported previously. Binarization classifies each pixel of the image as flow signal present or absent. Since choriocapillaris is originally a three-dimensional organization, a flow void is only be detected at a location where flow signal is absent throughout the entire slab thickness. Therefore, the number of flow voids in the current study might be underestimated. However, we could show flow void area was more in PPE eyes compared with age-matched normal controls.

Pachyvessels have been reported to be associated with inner choroidal attenuation [4, 20]. However, no previous studies have outlined the pathological causes underlying these findings. Takahashi K, et al. investigated the correlation between OCT and histopathological findings for pachychoroid spectrum disease and found that choroidal venous dilation and choriocapillaris dropout were involved in the pathological condition [30]. They hypothesized that the choriocapillaris flow deficit may be caused by compression of pachyvessels. A scanning electron micrograph study showed that the choriocapillaris comprises a single layer of capillaries constructed by lobular patterns with a terminal choroidal arteriole supplying independently an each lobule drained by a venule, and interarterial shunts occurred between two systems [31]. These shunts are considered to have the function of equalizing each arterial pressure and venous pressure and eliminating the difference in perfusion pressure between adjacent lobules. It is possible that the capillary lumen can be compressed by a dilated and congested vein. In the current study, we investigated the overlapping position between flow voids and pachyvessels. We showed that PPE eyes had more flow void area compared with normal eyes, and the mean proportion of the flow void area overlying the pachyvessels against the whole flow void area of the choriocapillaris was only 21.3%, suggesting that PPE eyes showed a choriocapillaris flow deficit regardless of pachyvessel location. These results suggest that the presence of pachyvessels did not directly cause choriocapillaris flow deficits. Gal-Or O and associates reported that reduced anatomical choriocapillaris zones do not necessarily correlate with pachyvessels and that inner choroidal ischemia seemed to involve the pathogenesis of pachychoroid diseases [14], which is consistent with our results. Taken together, the diffuse ischemic condition of choriocapillaris seems to be involved in the pathogenesis underlying pachychoroid spectrum diseases.

The precise mechanism underlying inner choroidal attenuation and ischemia in pachychoroid diseases remains unknown. Saito M, et al. hypothesized that in eyes with acute CSC, choroidal circulatory abnormalities resulted from increased sympathetic activity with arteriole vasoconstriction resulted in disturbed capillary perfusion [32]. Lee M, et al. proposed that choriocapillaris obliteration or ischemia may cause RPE damage leading to PPE [33]. Yanagi Y suggested that inner choroidal attenuation and ischemia of the choriocapillaris may subsequently upregulate angiogenic factors such as VEGF [34]. Further investigations are necessary to reveal the pathogenic mechanism underlying pachychoroid diseases.

Our study had several limitations. First, the number of patients was relatively small due to the rarity of PPE. Second, the choriocapillaris outside the 3 × 3 mm area was not evaluated. High-quality en face choriocapillaris images were obtained only with the 3 × 3 mm area size. Third, the current study showed no significant differences in number and total area and average size of flow voids between the pachyvessel and non-pachyvessel groups, which might be due to small patient number Fourth, the definition of pachyvessel is subjective and vague. No objective or quantitative definition has been established so far. In the current study, the pachyvesssel was diagnosed by two retinal specialists with multimodal images. The strength of this study was that it investigated the structure of choriocapillaris quantitively by using high-quality images obtained by multiple en face SS-OCTA averaging in PPE, which is thought to represent the earliest clinical stage of pachychoroid-related diseases.

In conclusion, the averaged en face OCTA images of the choriocapillaris revealed the microstructure of the choriocapillaris in detail. In PPE eyes, the blood flow area of the choriocapillaris decreased diffusely within the macular area in comparison with control eyes, and choriocapillaris flow deficit was not necessarily related to the location of pachyvessels. Our results provided additional evidence showing that blood flow disorders of choriocapillaris may be involved in the occurrence of pachychoroid spectrum diseases.

## Supporting information

**S1 Data.**
(XLSX)

## Author Contributions

**Conceptualization:** Sotaro Ooto.

**Data curation:** Miho Tagawa, Kenji Yamashiro, Hiroshi Tamura, Akio Oishi, Manabu Miyata, Masahiro Miyake, Ayako Takahashi, Ai Ichioka, Akitaka Tsujikawa.

**Formal analysis:** Miho Tagawa, Sotaro Ooto.

**Funding acquisition:** Sotaro Ooto.

**Investigation:** Miho Tagawa.

**Methodology:** Miho Tagawa, Sotaro Ooto, Akihito Uji.

**Software:** Akihito Uji.

**Supervision:** Sotaro Ooto, Akitaka Tsujikawa.

**Writing – original draft:** Miho Tagawa.

**Writing – review & editing:** Sotaro Ooto, Kenji Yamashiro, Hiroshi Tamura, Akio Oishi, Akihito Uji, Manabu Miyata, Masahiro Miyake, Ayako Takahashi, Ai Ichioka, Akitaka Tsujikawa.

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
