## [Decision Letter · Decision Letter 0]

10 Mar 2022

PONE-D-22-03740Choriocapillaris flow deficit in a pachychoroid spectrum disease using en face optical coherence tomography angiography averagingPLOS ONE

Dear Dr. Ooto,

Thank you for submitting your manuscript to PLOS ONE. After careful consideration, we feel that it has merit but does not fully meet PLOS ONE’s publication criteria as it currently stands. Therefore, we invite you to submit a revised version of the manuscript that addresses the points raised during the review process.

We look forward to receiving your revised manuscript.

Kind regards,

Ireneusz Grulkowski, PhD

Academic Editor

PLOS ONE

Journal Requirements:

- https://journals.plos.org/plosone/article?id=10.1371%2Fjournal.pone.0259880

- https://link.springer.com/article/10.1007/s00417-019-04399-8

In your revision ensure you cite all your sources (including your own works), and quote or rephrase any duplicated text outside the methods section. Further consideration is dependent on these concerns being addressed.

4. Thank you for stating the following in the Funding Section of your manuscript: 

"This study was supported in part by the Japan Society for the Promotion of Science (JSPS), Tokyo, Japan (Grant-in-Aid for Scientific Research, no. 16K11321). The funding body had no role in the design or conduct of the study, management, analysis, and interpretation of the data, or the preparation, review, or approval of the manuscript. "

"None of the authors has a proprietary interest in any of the products described herein."

"None of the authors has competing interests."

7. Please amend the manuscript submission data (via Edit Submission) to include author Akihito Uji.

Reviewers' comments:

Reviewer's Responses to Questions

**Comments to the Author**

1. Is the manuscript technically sound, and do the data support the conclusions?

Reviewer #1: Yes

Reviewer #2: Yes

2. Has the statistical analysis been performed appropriately and rigorously? 

Reviewer #1: Yes

Reviewer #2: Yes

3. Have the authors made all data underlying the findings in their manuscript fully available?

Reviewer #1: Yes

Reviewer #2: Yes

4. Is the manuscript presented in an intelligible fashion and written in standard English?

Reviewer #1: Yes

Reviewer #2: Yes

5. Review Comments to the Author

Reviewer #1: Major comments

Authors investigated the choriocapillaris changes associated with pachychoroid pigment epitheliopathy (PPE) in comparison with healthy eyes. Authors found that the blood flow area of the choriocapillaris decreased diffusely within the macular area in PPE eyes compared to control eyes, and the choriocapillaris flow deficit was not necessarily related to pachyvessel location.

This study shows that choriocapillaris changes in PPE eyes were not directly caused by the pachyvessels. This is an interesting finding.

The definition of pachyvessel is not clear. Authors stated that “Pachyvessels were defined as dilated choroidal venous veins that continuously extended from the dilated vortex vein, could be distinguished by their relatively larger caliber in comparison with other choroidal veins in other quadrants, and showed asymmetry of upper and lower vortex veins.” This is a subjective and vague definition, and it is difficult for other researchers to follow this study. This part should be stated in the limitation section.

In addition, this study does not pay attention to the distance between pachyvessel and PRE. Because PRE abnormality, PRE detachment, or choroidal neovascularization tend to occur on or near the site where this distance is very small, this distance is an important factor when evaluating PPE. This may be why there was no significant difference in Table 3. If authors ignored this parameter, this part also should be stated in the limitation section.

Minor comments

Eye of PPE were fellow eyes of CSC or normal eyes? Please add information of the patients of PPE.

Table 2 and Table 3: “p values adjusted for age and sex.”

Was this comparison possible using un-paired t-test, as stated in the Method.

Table 3: Please add the information of choroidal thickness.

Reviewer #2: Thank you for inviting me to review the manuscript entitled, "Choriocapillaris flow deficit in a pachychoroid spectrum disease using en face optical coherence tomography angiography averaging."

Comments:

1. This study included the eyes with PPE and nornal control eyes. However, the detailed information about the contralateral eyes were not provided. In terms of PPE, the contralateral eyes had no abnormalities or only PPE? If so, performing dye angiography sounds ethically problematic. Were there any exudative abnormalities in the contralateral eyes, such as CSC, PNV, pr PCV? In terms of nornal control eyes, were there any ocular diseases in the contralateral eyes? Furthermore, the status of contralateral eyes affected the results?

2. Please demonstrate more information about the systemic conditions that might influence the results (i.e, hypertension and diabetes) should be mentioned. Otherwise, some comments should be included as the limitation of the study.

End of comments.

6. PLOS authors have the option to publish the peer review history of their article (what does this mean?). If published, this will include your full peer review and any attached files.

Reviewer #1: No

Reviewer #2: No

---

## [Author Response · Author response to Decision Letter 0]

16 May 2022

Reviewer #1: Major comments

Authors investigated the choriocapillaris changes associated with pachychoroid pigment epitheliopathy (PPE) in comparison with healthy eyes. Authors found that the blood flow area of the choriocapillaris decreased diffusely within the macular area in PPE eyes compared to control eyes, and the choriocapillaris flow deficit was not necessarily related to pachyvessel location.

This study shows that choriocapillaris changes in PPE eyes were not directly caused by the pachyvessels. This is an interesting finding. The definition of pachyvessel is not clear. Authors stated that “Pachyvessels were defined as dilated choroidal venous veins that continuously extended from the dilated vortex vein, could be distinguished by their relatively larger caliber in comparison with other choroidal veins in other quadrants, and showed asymmetry of upper and lower vortex veins.” This is a subjective and vague definition, and it is difficult for other researchers to follow this study. This is part should be stated in the limitation section.

A. The presence of pachyvessels was evaluated by using choroidal wide-angle en face OCT images (12 × 12 mm) and wide-angle IA. The presence of pachyvessels in each case was determined in agreement by two retinal specialists. (line221)

As suggested by the reviewer, we added the followings in the limitation section.

“Fourth, the definition of pachyvessel is subjective and vague. No objective or quantitative definition has been established so far. In the current study, the pachyvesssel was diagnosed by two retinal specialists with multimodal images.” (line 443)

In addition, this study does not pay attention to the distance between pachyvessel and PRE. Because PRE abnormality, PRE detachment, or choroidal neovascularization tend to occur on or near the site where this distance is very small, this distance is an important factor when evaluating PPE. This may be why there was no significant difference in Table 3. If authors ignored this parameter, this part also should be stated in the limitation section.

A. As the reviewer suggested, we mentioned the parameter and added or revised the following in the methods, Table3, Results, and limitation section, respectively.

“Measurement of the distance between RPE and choroidal venous vein

We evaluated the distance between RPE and choroidal venous vein. We manually measured the distance at the center of the fovea, 1mm superior side from the fovea, 1mm inferior side from the fovea, 1mm nasal side from the fovea, and 1mm temporal side from the fovea by using a built-in caliber tool. We used vertical and horizontal B-scans through the fovea, and averaged each distance.” (line 240)

“The two groups showed no significant differences in number of flow voids, total flow void area, average size of the flow voids, and SFCT after adjusting for age and sex. The distance between RPE and choroidal venous vein in pachyessel group was significantly shorter than that in non-pachyvessel group.” (line 318)

Minor comments

1. Eye of PPE were fellow eyes of CSC or normal eyes? Please add information of the patients of PPE.

A. As suggested by the reviewer, we added the following in methods section.

“Eyes of PPE were fellow eyes of PNV, CSC or PPE. If both eyes were eligible, the right eye was included.” (line 153)

2. Table 2 and Table 3: “p values adjusted for age and sex.”

Was this comparison possible using un-paired t-test, as stated in the Method.

A. Thank you for helpful advice. We used partial correlation for adjusting for age and sex. The followings were added in statistical analysis in methods section.

“Partial correlation was used for adjusting for age and sex.” (line 263)

3. Table 3: Please add the information of choroidal thickness.

A. We added the information of choroidal thickness in Table 3. 

Reviewer #2: Thank you for inviting me to review the manuscript entitled, "Choriocapillaris flow deficit in a pachychoroid spectrum disease using en face optical coherence tomography angiography averaging."

Comments:

1. This study included the eyes with PPE and normal control eyes. However, the detailed information about the contralateral eyes were not provided. In terms of PPE, the contralateral eyes had no abnormalities or only PPE? If so, performing dye angiography sounds ethically problematic. Were there any exudative abnormalities in the contralateral eyes, such as CSC, PNV, pr PCV? In terms of normal control eyes, were there any ocular diseases in the contralateral eyes? Furthermore, the status of contralateral eyes affected the results?

A. As the reviewer suggested, we provided the detailed information about the contralateral eyes of PPE eyes. The following were added in participants in methods section.

“Eyes of PPE were fellow eyes of PNV, CSC or PPE. If both eyes were eligible, the right eye was included.” (line 153)

We mentioned about the information of normal control eyes in participants in methods section. 

“We recruited healthy eyes of healthy volunteers and unaffected fellow eyes of patients with unilateral retinal diseases (e.g., epiretinal membrane, vitreomacular traction syndrome) [16]. In the current study, subjects with the systemic conditions like diabetes mellitus or those with poorly controlled hypertension were excluded.” (line 154)

We analyzed whether the status of contralateral eyes affected the results. 

“Fellow eyes of eyes with PPE were PNV: 17 eyes, CSC: 11 eyes, PPE: 4 eyes.

In PNV eyes, the mean number of flow voids were significantly smaller than those in CSC eyes (1432 ± 147 vs 1592 ± 111; p = 0.018). In PNV eyes, the mean average size of the flow voids was significantly larger than those in CSC eyes (842 ± 144 vs 704 ± 76; p = 0.041). However, there were no significant differences in the total flow void area among 3 groups.” (line308)

2. Please demonstrate more information about the systemic conditions that might influence the results (i.e, hypertension and diabetes) should be mentioned. Otherwise, some comments should be included as the limitation of the study.

A. As the reviewer suggested, we added the followings in methods section. 

“In the current study, subjects with the systemic conditions like diabetes mellitus or those with poorly controlled hypertension were excluded.” (line 156)

---

## [Decision Letter · Decision Letter 1]

7 Jul 2022

Choriocapillaris flow deficit in a pachychoroid spectrum disease using en face optical coherence tomography angiography averaging

PONE-D-22-03740R1

Dear Dr. Ooto,

We’re pleased to inform you that your manuscript has been judged scientifically suitable for publication and will be formally accepted for publication once it meets all outstanding technical requirements.

Kind regards,

Ireneusz Grulkowski, PhD

Academic Editor

PLOS ONE

Additional Editor Comments (optional):

Reviewers' comments:

Reviewer's Responses to Questions

**Comments to the Author**

1. If the authors have adequately addressed your comments raised in a previous round of review and you feel that this manuscript is now acceptable for publication, you may indicate that here to bypass the “Comments to the Author” section, enter your conflict of interest statement in the “Confidential to Editor” section, and submit your "Accept" recommendation.

Reviewer #2: All comments have been addressed

2. Is the manuscript technically sound, and do the data support the conclusions?

Reviewer #2: Yes

3. Has the statistical analysis been performed appropriately and rigorously? 

Reviewer #2: Yes

4. Have the authors made all data underlying the findings in their manuscript fully available?

Reviewer #2: Yes

5. Is the manuscript presented in an intelligible fashion and written in standard English?

Reviewer #2: Yes

6. Review Comments to the Author

Reviewer #2: Thank you for allowing me to review the manuscript. The authors have adequately addressed my comments raised in a previous round of review.

7. PLOS authors have the option to publish the peer review history of their article (what does this mean?). If published, this will include your full peer review and any attached files.

Reviewer #2: No

---

## [Editor Report · Acceptance letter]

2 Sep 2022

PONE-D-22-03740R1 

Choriocapillaris flow deficit in a pachychoroid spectrum disease using en face optical coherence tomography angiography averaging 

Dear Dr. Ooto:

I'm pleased to inform you that your manuscript has been deemed suitable for publication in PLOS ONE. Congratulations! Your manuscript is now with our production department. 

Kind regards, 

on behalf of

Dr. Ireneusz Grulkowski 

Academic Editor

PLOS ONE